# Why Do We Need Alternative Methods for Fungal Disease Management in Plants?

**DOI:** 10.3390/plants12223822

**Published:** 2023-11-10

**Authors:** Michael S. McLaughlin, Maria Roy, Pervaiz A. Abbasi, Odile Carisse, Svetlana N. Yurgel, Shawkat Ali

**Affiliations:** 1Agriculture and Agri-Food Canada, Kentville Research and Development Centre, Kentville, NS B4N 1J5, Canada; shayne.mclaughlin@agr.gc.ca (M.S.M.); maria.roy@agr.gc.ca (M.R.); p.abbasi@hotmail.com (P.A.A.); 2Department of Plant, Food and Environmental Sciences, Faculty of Agriculture, Dalhousie University, Truro, NS B2N 4H5, Canada; 3Department of Biology, Acadia University, Wolfville, NS B4P 2R6, Canada; 4Saint-Jean-sur-Richelieu Research Development Centre, Science and Technology Branch, Agriculture and Agri-Food Canada, Saint-Jean-sur-Richelieu, QC J3B 7B5, Canada; odile.carisse@agr.gc.ca; 5United States Department of Agriculture (USDA), Agricultural Research Service, Grain Legume Genetics and Physiology Research Unit, Prosser, WA 99350, USA; svetlana.yurgel@usda.gov

**Keywords:** plant defence elicitors, biological control, biochemical fungicides, RNA interference, fungal disease management

## Abstract

Fungal pathogens pose a major threat to food production worldwide. Traditionally, chemical fungicides have been the primary means of controlling these pathogens, but many of these fungicides have recently come under increased scrutiny due to their negative effects on the health of humans, animals, and the environment. Furthermore, the use of chemical fungicides can result in the development of resistance in populations of phytopathogenic fungi. Therefore, new environmentally friendly alternatives that provide adequate levels of disease control are needed to replace chemical fungicides—if not completely, then at least partially. A number of alternatives to conventional chemical fungicides have been developed, including plant defence elicitors (PDEs); biological control agents (fungi, bacteria, and mycoviruses), either alone or as consortia; biochemical fungicides; natural products; RNA interference (RNAi) methods; and resistance breeding. This article reviews the conventional and alternative methods available to manage fungal pathogens, discusses their strengths and weaknesses, and identifies potential areas for future research.

## 1. Introduction

Plant pathogens pose a significant threat to the agricultural industry and are one of the most important factors in agricultural yield losses and food insecurity across the globe. Fungal pathogens alone may account for up to 20% of worldwide yield losses [1]. Conventional breeding techniques alone cannot provide adequate protection against fungal pathogens for many crops because fungal pathogens are able to overcome introduced genetic resistance [2]. These pathogens pose a particularly serious problem in perennial crops such as apples and other tree fruits. Orchards are expected to last 20–30 years, making it unrealistic to replace vulnerable cultivars with resistant ones, especially since resistance can be overcome long before the end of an orchard’s productive life. Considering that genetic resistance is generally not sustainable and the development of resistant cultivars takes many years, disease control has relied for decades on the application of chemical fungicides [2]. Although very effective, these fungicides are notorious for their hazardous effects on human and animal health as well as for their environmental toxicity [3,4]. Concerns over the potential environmental consequences of the uncontrolled use of these active substances has led to regulations on their use based on environmental risks assessments, and these restrictions can range from reductions in the number of applications per crop season to the outright removal of specific active substances from the market [3]. In addition, plant pathogenic fungi can develop resistance to chemical fungicides, particularly single-site fungicides, which are more likely to lead to the development of resistance in fungal pathogen populations. In the past few decades, numerous disease management strategies have been developed as an alternative to traditional chemical fungicides and breeding methods, including the use of plant defence elicitors (PDEs), biological control or biochemical fungicides, and RNA interference (RNAi). We have reached a crossroads in which these alternatives to chemical fungicides will be called on to play an increasingly important role in disease management. The present article summarizes the current state of fungal disease management strategies, discusses the strengths and weaknesses of their modes of action, and draws conclusions on the future of fungal disease management.

### 1.1. Chemical Fungicides

Fungicides can be broadly defined as chemical substances used to control fungal diseases by inhibiting the growth of pathogenic fungi or by eradicating them completely. Fungicides can be classified as inorganic or organic based on their chemical composition. Inorganic fungicides do not contain carbon in their structure and are typically based on sulphur or metal ions. This group of fungicides has been in use since the discovery of Bordeaux mixture (copper sulphate pentahydrate and lime) by Pierre-Marie Alexis Millardet in the in the late 19th century [5]. Centuries after this discovery, copper- and sulphur-based fungicides are still used extensively in conventional and organic agriculture [6,7,8,9]. Examples of modern inorganic fungicides include copper sulphate, copper oxychloride, and copper hydroxide [10,11]. In contrast, organic fungicides contain carbon atoms in their structure [6]. These organic synthetic compounds have become more popular, although inorganic fungicides continue to be used in modern agriculture. All synthetic inorganic or organic fungicides, regardless of their composition, can be divided into two distinct classes based on their mobility in the plant: contact (protective) fungicides, which remain on the surface of the plant, and systemic (mobile/curative) fungicides, which are absorbed into the plant.

Contact fungicides typically have a wide range of action against different fungal pathogens and are effective in preventing the infection of plant tissues. They usually achieve this by killing fungal spores or by inhibiting their germination before they penetrate and colonize the host tissues [6,12]. Contact fungicides are not effective in a curative strategy and to be effective, must be applied before the pathogen infects the plant [13]. Most contact fungicides are not absorbed by the plant and remain on the plant tissue surface. However, the frequency of application must be carefully monitored, since contact fungicides can become phytotoxic in the rare cases when they are absorbed [14]. Because contact fungicides usually remain on the plant surface, protection is temporary and can be quickly lost due to rainfall or other weather conditions [15,16]. Contact fungicides can also be eliminated from the plant surface by wind or degraded by UV radiation, and therefore, their protective action does not exceed 10–12 days [6]. It is also important to bear in mind that contact fungicides are only effective on the leaf surface present at the time of application and thus are less efficacious during heavy leaf growth. Therefore, the effectiveness of contact fungicides is particularly reliant on the proper timing of application, which generally must be performed prior to the known or predicted infection periods of the targeted pathogens. Furthermore, since these fungicides are gradually removed from the plant surface, repeated applications during the growing season are necessary for sustained protection. In some circumstances, this characteristic is beneficial to growers, since contact fungicides, unlike systemic fungicides, have the advantage of being easily removed from treated produce before it reaches the consumer [6].

Systemic fungicides are a more recent development in disease control and are considered to be more promising than contact fungicides [17]. While providing a protective effect by suppressing spore germination, these fungicides can also be absorbed into plant tissues, either locally or more broadly, and are therefore able to kill fungal pathogens after they have penetrated and infected host tissues [18]. The degree of systemic activity—which ranges from simple translaminar activity in leaf tissues to local spread from the absorption site and mobility within the xylem of the plant—is generally determined by the chemistry of the compound and can play an important role in determining the efficacy of a fungicide against specific fungal pathogens [18,19,20]. Because of the ability of systemic fungicides to be absorbed in plant tissues, using them to treat plant materials has become routine practice, and the seeds of most agricultural crops are treated with systemic fungicides to protect against both seed- and soil-borne pathogens [21]. Although systemic fungicides are highly effective, most of the compounds involved operate through a single mode of action (i.e., they generally target a single essential fungal enzyme or metabolic pathway) and, therefore, are extremely vulnerable to the development of resistance by target fungal pathogens [22].

### 1.2. The Disadvantages of Chemical Fungicides: Environmental Toxicity and Resistance Development

Despite their high efficacy, both contact and systemic fungicides have numerous drawbacks associated with their use. One substantial shortcoming is that, due to their lack of specificity, chemical fungicides can disrupt both beneficial and pathogenic microorganisms. For instance, the application of fungicides to mango leaves has been demonstrated to eliminate many endophytes, creating a window of opportunity for pathogens to colonize the tissues formerly that the endophytes formerly inhabited [23]. While this phenomenon has been most readily observed in foliar spray treatments, fungicidal seed treatments have also been associated with similar reductions in beneficial endophytes, and the negative consequences of fungicides on soil microbial communities are well documented [24,25,26,27,28]. Indeed, mounting evidence suggests that seeds protected with systemic fungicides may negatively impact plant health and vigour by targeting beneficial endophytes in the absence of pathogen pressure, the practice may be counterproductive [21]. Therefore, the application of chemical fungicides can have negative consequences for plant health and yield by eliminating beneficial microbes that promote growth, development, and resistance to biotic and abiotic stresses.

Fungicides can severely impact the aquatic environment, as they are able to enter aquatic ecosystems through different ways, including wastewater, runoff, and subsurface drainage [29,30], and can be toxic to a wide range of aquatic organisms, including algae, fish, and invertebrates [31,32]. Furthermore, fungicides could harm important pollinators like bees through mechanisms such as the impairment of larval and physiological development, the promotion of increased sensitivity to other pesticides, and increased mortality [33]. Exposure to fungicides can lead to acute and chronic neurotoxicity in humans, and thus significantly impact human health [34].

In addition to fungicides’ environmental toxicity, concerns have been raised over the durability of fungicide efficacy. Fungi have tremendous evolutionary potential to rapidly develop resistance against fungicides due to the intense selective pressure exerted by repeated fungicide applications [22]. Mutations in DNA sequences can arise from errors in DNA replication, damage from UV radiation, or exposure to mutagens or viral infections. Environmental stress may play a significant role in determining the rate of mutation. For example, stress from increasing temperatures accelerated the rate observed in the fungal pathogen *Zymoseptoria tritici* [35]. Although mutations are inherently random, those that result in enhanced resistance to fungicides will be positively selected for by the eradication of strains without resistance. Over time, this inevitably results in fungicide-resistant strains of the targeted pathogen [22].

Fungicide resistance is a stable and heritable change in an individual fungus that results in a reduction in its susceptibility to fungicides. Fungicide resistance is well documented to develop more often against single-site fungicides than those with a multi-site mode of action, making modern synthetic fungicides especially vulnerable to resistance development [22]. The threat of fungicide resistance is a major concern to growers worldwide, and numerous strategies are employed in order to prevent its emergence. The Fungicide Resistance Action Committee broadly divides fungicides by their mode of action in order to identify those with potential resistance concerns. Its recommendations include applying multiple fungicides with varying modes of action over the course of the growing season, restricting the use of the fungicides most likely to induce resistance with repeated use, and prioritizing multi-site fungicides, to which fungi are less likely to develop resistance [36]. To date, over 43 different modes of action have been identified, although the mechanisms for some of these are not yet known [36].

In conclusion, while chemical fungicides are an extremely effective tool—at least in the short term—for reducing disease incidence in the crops, they have harmful effects on beneficial plant microbiota, the health of humans and other animals, and on the environment. These factors, in addition to the rising threat of fungicide resistance, have led to increasing restrictions on the use of chemical fungicides. Given these challenges, conventional chemical fungicides must be complemented with cost-effective, eco-friendly alternatives to maintain appropriate levels of disease control with the absence or reduced usage of these vital compounds. An overview of the benefits and drawbacks of chemical fungicides and alternative disease control methods can be found in Figure 1.

## 2. Alternative Management of Fungal Diseases

### 2.1. Agronomic Practices and Cultivation Methods

Agronomic practices and cultivation methods can greatly influence the vulnerability of crops to fungal pathogens through a variety of mechanisms, including disruption of the pathogen lifecycle, improving the vegetative performance and thus the natural health status of the plant host, or the removal of sources of inoculum from the field. The main drawback of these agronomic practices is that they tend to be laborious, expensive, and less effective than conventional fungicides, and thus, they are more suited as complementary tools to more effective methods. These practices include but are not limited to: sanitation, tillage, crop rotation, pruning and thinning, the intermixing of different crops or different varieties of the same crop, and the manipulation of canopy architecture. The impacts of cultivation methods on pathogen pressures have been the subject of many thorough reviews [37,38,39], and thus, these practices will only be briefly described in this review.

Foremost among cultivation methods which can enhance the control of fungal pathogens is proper sanitation, that is, the removal of sources of inoculum from the field. For instance, the removal of crop residues, which often serve as the source of primary inoculum for *Colletotrichum* species, is an effective means for reducing anthracnose and black spot in guava fruit [40]. Leaf shredding is a common means of removing inoculum sources of foliar pathogens and has long been promoted as a means to control apple scab (*Venturia inaequalis*) [41]. However, as with many cultivation practices, sanitation practices on their own do not provide disease control comparable to chemical fungicides. In a direct comparison, the removal of senescent and necrotic leaves and the removal of unmarketable fruit from the alleys between beds of strawberries significantly reduced Botrytis fruit rot caused by *Botrytis cinerea* in annual strawberry compared to controls, but losses remained significantly higher than in the fungicide control [42]. Nevertheless, sanitation practices are an effective means of complementing chemical fungicides. Recently, it has been demonstrated that leaf shredding in combination with the application of demethylation inhibitor fungicides significantly influenced the fungicide sensitivity of *V. inaequalis* populations. *V. inaequalis* isolates from orchards treated with demethylation inhibitor fungicides which also underwent leaf shredding retained fungicide sensitivities close to that of unexposed populations, potentially as a result of a smaller initial effective population size [43]. Thus, the inclusion of proper sanitation practices with chemical fungicides in an integrated treatment regime may both improve disease control and delay the development of fungal resistance, though further research will be required to determine the extent of this effect.

There is conflicting evidence on the impacts of tillage on disease control, and whether specific tillage practices promote or reduce fungal diseases is reliant on the specific pathogen and macro-environmental conditions [44,45,46]. The removal of organic matter in conventional tillage practices, while consequential for soil health, also removes potential sources of primary inoculum, while conservation (or reduced) tillage can promote pathogen survival by providing residues for the pathogen between crop plantings [47]. Furthermore, conservation tillage can alter soil characteristics in ways which may either be beneficial or detrimental to individual pathogens, such as increasing soil moisture, altering soil temperatures, and failing to disrupt the soil [48]. However, compared to conventional tillage, conservation or no-tillage has been shown to improve the general disease suppressiveness of the soil by reducing tillage-induced losses in microbial biomass, and disease suppression in spring barley was improved by long-term conventional tillage and no-tillage [46]. Given the significance of the soil microbiome in the determination of the prevalence of disease, particularly in the case of soil-borne pathogens, the impact of these practices on soil and rhizosphere microbiomes is an area of intense interest [49,50].

The continuous cultivation of a particular crop results in the accumulation of pathogens and increased disease pressure [51]. As such, the rotation of crops from host to non-host species serves as an effective break to reduce pathogen inoculum [52]. Crop rotation is especially important for the control of soil-borne pathogens, and longer duration periods have been demonstrated to be more effective in preventing disease [53]. Furthermore, crop rotation may have beneficial effects on the general disease suppressiveness of the soil. Numerous studies have investigated crop rotation either alone or in combination with tillage for its potential in modulating general disease suppressiveness in the soil in crop systems such as potato, banana, and peanut [53,54,55]. Thus, ideal rotation strategies will not only serve as a break in pathogen pressure but as a means of enriching the soil’s capacity to ward off pathogens. Similar to crop rotations, the inter-cropping of different species can serve to reduce pathogen pressures [56]. In this respect, the examination of soil microbiomes as a result of multi-cropping in five organic vegetable farming systems revealed that inter-cropping was associated with a decrease in the abundance of soil-borne pathogens [57]. Thus, determining the ideal combinations for inter-cropping for each species may serve as an effective form of disease control.

Plant canopy architecture is another important factor in determining disease pressure, as the density of the canopy can significantly influence the micro-climate of the canopy as well as spore dispersal. Thus, for fungal pathogens that are dispersed over short distances by splashing rain, canopy architecture is a major factor in determining the dispersal of the pathogen [58,59,60]. On the other hand, dense canopies are also associated with a microclimate of increased humidity which favours pathogen infection, and thinning systems which promote aeration within the canopy have been shown to reduce the incidence of apple scab infection [61,62]. Thus, the determination of the optimal density for a given crop represents a significant challenge in the pursuit of enhanced disease control. Intriguingly, the uniformity of crop height in current cropping systems may be advantageous to splash dispersed fungal pathogens, as it has recently been demonstrated that growing wheat cultivars of contrasting height together enhanced the control of *Septoria triciti* blotch (*Zymoseptoria triciti*) [63]. While more research is necessary to determine whether this approach is applicable to other crop systems or pathogens, it is clear that further research into the interplay between canopy architecture and disease will be beneficial for disease management.

Whether it be through the direct removal of primary inoculum sources, the rotation of non-host crops, or the establishment of physical barriers to pathogen dispersal, cultivation methods have significant influence on the vulnerability of crops to fungal pathogens. Understanding how common cultivation practices interact with disease management will allow for the optimization of growth conditions in order to reduce disease pressure, especially as these cultivation practices can readily be combined with chemical fungicides or their alternatives for improved disease control.

### 2.2. Improving Plants’ Genetic Resistance through the Use of R and S Genes

Plants have evolved numerous genetic defence mechanisms to protect themselves from pathogens. Growers have long relied on the manipulation of these mechanisms, traditionally by breeding for resistance as a way to reduce crops’ susceptibility to fungal pathogens. Host plants can recognize non-specialized fungal pathogens by toll-like receptors that detect pathogen-associated molecular patterns, in turn activating the host’s downstream defense mechanisms. Specialized pathogens are able to overcome these basal defence mechanisms by secreting effector molecules into host plants [64]. However, the co-evolution of plants and fungal pathogens over millennia has provided plants with a means of defence against effector molecules: resistance (R) genes. The R-gene family is incredibly diverse and well conserved in plant species. R-genes encode for nucleotide-binding leucine-rich receptors (NLRs), which collectively recognize a broad spectrum of plant pathogens and pests, inducing an array of resistance mechanisms in response to infection or predation [65,66]. NLRs are activated by the binding and recognition of pathogen effectors but, in some cases, may detect a pathogen indirectly, generally by recognizing pathogen modified host proteins. An example of this form of recognition occurs in *Arabidopsis* in response to *Pseudomonas syringae* infection, where the effector (in this case, a protease) cleaves the *Arabidopsis* PBS1 kinase, triggering its recognition by the NLR RPS5 [67]. The successful recognition of effector molecules or effector-modified host proteins in plants by the associated NLR typically results in effector-triggered immunity, a localized response characterized by a hypersensitive reaction (HR) in which the plant cells near the infection undergo apoptosis [68,69]. In addition, the recognition of a pathogen by plant pattern-recognition receptors (PAMPs) or an effector protein by R proteins triggers the production of salicylic acid (SA) and the downstream induction of broad, systemic defence mechanisms against subsequent infections, triggered independently of the HR response [70,71].

The direct or indirect recognition of effector proteins by *R*-gene-encoded receptors involves a gene-for-gene relationship in which the *R*-gene receptor identifies a single effector protein (encoded by a matching avirulence [*Avr*] gene); therefore, a host with a given *R*-gene will be resistant to a pathogen with the matching *Avr* gene [72]. In most plant–pathogen systems, the host and pathogen species may collectively have numerous *R*- or *Avr* genes. For example, twenty *R*-genes have been identified in apples (*Malus* × *domestica*) that match the corresponding *Avr* genes identified in the pathogen *Venturia inaequalis*, which causes apple scab. However, it should be noted that no single cultivar or individual line will contain all these resistance genes; for example, Honeycrisp apples have *Rvi19* and *Rvi20* in their genomes, while Golden Delicious cultivars contain *Rvi1*. Therefore, while many apple cultivars have some resistance to *Venturia inaequalis*, these cultivars are still vulnerable to some *Venturia inaequalis* strains that do not have corresponding *Avr* genes [73].

A typical mechanism in pathogens for overcoming host resistance is *Avr* gene mutations to prevent the product (or activity, in the case of indirect mechanisms) from being recognized by *R*-gene-encoded receptors. If the effector is recognized, pathogens can also overcome resistance by interfering with the host response [74]. The presence of an *R*-gene in a host plant population will naturally select for pathogens in which the corresponding *Avr* gene has been lost or modified so that it is no longer recognized by the *R*-gene-encoded receptor. In turn, successful mutations in the *Avr* gene will induce selection pressure on host plants for *R*-genes which impart resistance to the mutated effector. Thus, host plants and their pathogens are continuously engaged in an evolutionary arms race and, in wild populations, the frequencies of *Avr* and *R*-genes will cycle over time [75,76]. In modern agricultural settings, the uniformity of resistance genes in a population may accelerate the selection process, leading to rapid loss of resistance in these settings [77].

The identification of *R*-genes and their incorporation in economically important crops is a vital pillar in the development of resistant plants. Along with the use of conventional fungicides, resistance breeding techniques have served as the most effective method of disease control for decades, particularly in annual crops [78]. Although resistance breeding is also practiced in perennial crops, its effectiveness is often limited by the lifetime of the crop. Modern tree fruit crops, for instance, are expected to have lifespans of 20–30 years, giving ample time for selective pressure from resistant cultivars to result in pathogens overcoming the associated *R*-genes [77]. This is particularly problematic since introducing cultivars with new resistance genes is difficult due to orchards’ long lifespans. Furthermore, plant breeding is very time-consuming and, in recent years, plant breeders have relied on transgenic tools or gene transformation to expedite resistance-breeding efforts, since they allow the faster introduction of *R*-genes from otherwise incompatible species as well as from compatible species [79].

Numerous strategies have been developed to reduce the ability of fungal pathogens to overcome *R*-genes, such as rotating *R*-genes in a field (most suitable for annual crops), mixing cultivars with distinct *R*-genes in a field or between fields, and pyramiding multiple *R*-genes in a single cultivar to confer more durable resistance [77]. Somewhat like multi-site fungicides, pyramiding *R*-genes in a single cultivar makes it more difficult for pathogens to overcome resistance despite their evolutionary potential [80]. However, pathogens are still capable of overcoming multiple *R*-genes in the same host plant. For example, the oomycete pathogen *Phytophthora infestans* can escape multiple resistance genes in potato [81]. The breakdown of resistance to rust fungi in cereal crops under different strategies was recently modelled, and it was demonstrated that, although pyramiding could provide the most effective pathogen resistance, this resistance is less durable when mutation rates in the pathogen population are high [77]. Under such conditions, mixing or rotating crops was more successful at delaying the breakdown of resistance to different *R*-genes. For example, in mixed populations, the breakdown of resistance to one major *R*-gene was correlated with increased durability of the other *R*-genes in the population. Rotations were particularly successful since they were modelled so that pathogens were consistently challenged with new *R*-genes. Consequently, the authors concluded that rotating different pyramids of *R*-genes was the most promising method of ensuring durable *R*-gene resistance [77]. However, many resistant varieties may incur yield and/or crop quality penalties when compared to their susceptible counterparts, and these costs must be carefully considered with the associated benefits [82]. Therefore, the rotation of resistant varieties may not be a feasible strategy in many cases due to economic implications. Furthermore, this strategy is unlikely to be useable or effective in perennial crops, such as apples, pears, and cherries, which have longer lifespans and a juvenile period.

Beyond the introduction of *R*-genes in susceptible genotypes, advances in genome editing have allowed researchers to identify other mechanisms for reducing disease severity or improving resistance, such as targeting susceptibility (*S*) genes [83]. *S*-genes are genes in the host plant required for pathogen infection. Interaction of a pathogen’s effector/toxin molecules with *S*-genes can assist the pathogen in a variety of ways, such as the recognition and penetration of the host, sustained compatibility between the pathogen and host, and the inhibition of immune signalling [84]. Therefore, the genetic silencing or knocking out of *S*-genes can improve the host plant’s resistance to the pathogen and is one of the newest frontiers in conferring durable pathogen resistance [83]. Recently, CRISPR/Cas9-mediated knockouts of three *S*-genes in potato, *StDND1*, *StCHL1* and *StDMR6*, increased resistance to potato late blight caused by *Phytophthora infestans* [85]. Likewise, in apple, the expression of the *MdCNGC2* gene, which encodes a cyclic nucleotide-gated ion channel, was observed to be strongly induced by *Botryosphaeria dothidea* infection in susceptible cultivars [86]. Improved resistance to the pathogen was observed with both virus-induced gene silencing and CRISPR/Cas9-mediated mutagenesis of *MdCNGC2* [86]. To date, targeting *S*-genes has proven to be a successful strategy for inducing disease resistance in a number of crop systems, including cucumbers, rice and tomato [87].

Directly introducing resistance in crops is an effective disease management strategy. However, while both *R*-genes and *S*-genes can be modified or integrated in the host genome to improve disease resistance, the process is costly, laborious, and time-consuming. In addition, the rapid breakdown of resistance in the field makes resistant cultivars less effective in long-lived crops. Therefore, complementary tools are needed to help delay the breakdown of resistance in crops that cannot be rotated annually.

## 3. The Use of Plant Defence Elicitors

While pathogen-triggered immunity coordinates the host’s defence against specific pathogens, systemic acquired resistance (SAR) or induced systemic resistance (ISR) mediates prolonged, broad-range resistance to plant pathogens [88,89]. The induction of SAR is characterized by a local increase in levels of the phytohormone SA, which in turn results in the accumulation of reactive oxygen species (ROS) and promotes the expression of β1,3-glucanase, chitinase, and classical pathogenesis-related (PR) genes (*Pr1*, *Pr2*, and *Pr5).* The increased expression of these genes promotes plant defence through various downstream targets [90]. Pathogen infection is associated with a rapid increase in SA levels in apple leaf tissue, and this activity is necessary to induce SAR [91,92]. ISR, however, does not require SA accumulation and instead is associated with the accumulation of jasmonic acid (JA) and ethylene (ET). JA and ET accumulation leads to the activation of downstream targets, including chitinase, β1,3-glucanase, and an alternative set of PR genes (*PDF1.2*, *PR3*, and *PR4*), as well as the accumulation of ROS (like in SAR) [93]. Significant overlap and crosstalk occur between the SAR and ISR signalling pathways, and both are associated with protection against different pathogens. For example, SAR provides greater protection against biotrophic or hemi-biotrophic pathogens but may leave the plant more vulnerable to necrotrophic pathogens. Conversely, ISR provides protection against necrotrophic pathogens and chewing insects at the expense of protection against biotrophs and hemi-biotrophs [94,95]. This is a direct result of the crosstalk between these pathways, since the upregulation of SA is typically associated with a decrease in JA and vice versa, although synergism occurs between these two defence mechanisms in some cases [95].

Inducing plant defence by applying exogenous elicitors is a promising alternative to conventional fungicide applications [96]. The application of exogenous PDEs has been associated with enhanced resistance against a wide range of pathogens in different crops, including cereals, tomato, rubber tree, and apple [97,98,99,100]. Pre-treatment with SA of the leaves of the susceptible apple cultivar Gala was associated with a marked (albeit temporary) increase in the expression of the plant pathogenesis-related genes *PR1*, *PR5* and *PR8*, as well as of chitinase and β1,3-glucanase, with the treated leaves demonstrating increased resistance to Glomerella leaf spot following in vitro inoculation with *Glomerella cingulate*, indicating the induction of SAR [100]. Similarly, exogenous applications of phenylacetic acid have been observed to induce ISR in tobacco, imparting significant resistance to the bacterial soft rot pathogen *Pectobacterium carotovum* [101]. Elicitor treatment may protect fruit from plant pathogens for a significantly longer period than the temporary uptick in defence-gene-related expression suggests. In this regard, field treatments of Ya Li pear with a 2.5 mM SA spray produced a remarkable decline in disease incidence and lesion diameter compared to the untreated control. Finally, activities of defence-related enzymes such as peroxidase, phenylalanine ammonia-lyase, chitinase, and β-1,3 glucanase not only increased significantly in the four days following SA application, but these increased activities were still observed after harvest [102].

Since the discovery of SA as an exogenous inducer of SAR, several synthetic PDEs that are analogues of the phytohormones SA and JA have been developed. The ones most commonly used in the past few decades are benzo(1,2,3)thiadiazole-7-carbothioic acid S-methyl ester (BTH) and 2,6-dichloro-isonicotonic acid (INA), which are preferred for their increased efficiency and reduced phytotoxicity compared to the original SA [90].

Phytohormones and their synthetic analogues are not the only methods for inducing plant defence mechanisms. Peptides, polysaccharides, and lipids isolated from plants, fungi, and bacteria can also serve as PDEs. For example, chitosan, which can be readily obtained from fungal cell walls, is well known for its ability to induce host defenses, although this compound also demonstrates direct fungicidal activity [103]. The application of ZhiNenCong (ZNC), an extract of *Paecilomyces variotii*, stimulated immunity in *Arabidopsis thaliana* to bacterial infection and in potato to *Phytophthora infestans* infection via SA-dependent signalling pathways. Cell wall extract from the mushroom *Pleurotus ostreatus* enhanced defence against *Septoria* and mildew in wheat and grapevine, respectively [104,105]. Similarly, extracts of giant knotweed (*Reynoutria sachalinensis)* improved the resistance of courgette to *Podosphaera xanthii* in an SA-dependent manner [106]. Extracts from the seaweeds *Ascophyllum nodosum*, *Cystoseira myriophylloides*, *Laminaria digitata*, and *Fucus spiralis* represent a significant portion of plant-based elicitors [107,108].

The application of exogenous plant defence elicitors that can stimulate SAR or ISR responses before pathogen infection is widely considered to be an eco-friendlier alternative for disease control in plants, and therefore, the identification and characterization of plant defense elicitors have been major areas of study (Table 1) [109]. However, PDEs have been shown to be significantly less effective than conventional chemical fungicides in many instances [102], which could be partially attributed to the inherent crosstalk between the SAR and ISR signalling pathways, since the induction of one pathway is often associated with the inhibition of the other. Therefore, the activation of SAR may improve a host’s defence against biotrophic and hemi-biotrophic fungal pathogens but leave the host more vulnerable to necrotrophs [95]. Furthermore, the induction of plant defences requires the significant allocation of the host plant’s resources and, thus, may be associated with a decrease in overall plant fitness [110,111]. For this reason, many PDEs are very rarely used, and therefore do not fully replace conventional chemical fungicides. Conversely, the use of elicitor products improved retention and yield, particularly in the case of seaweed extracts, which may also act as biostimulants [112,113]. These products can improve disease protection when applied in conjunction with fungicides. For example, the use of Actigard^®^ (an SA analogue) increased the efficacy of dimethomorph, mancozeb, and azoxystrobin in preventing tobacco blue mould caused by *Peronospora tabacina* [114]. Consequently, despite their limitations, PDEs remain an attractive disease control tool, with their usefulness determined by the context of their use. Although PDEs are inefficient when used on their own, when incorporated in integrated disease management programs, they could potentially reduce the use of chemical fungicides, thus lessening the environmental exposure to these pesticides and potentially slowing the development of fungicide resistance in pathogens. The increased use of PDEs combined with biological control agents or biochemical fungicides could provide an economically acceptable level of control.

## 4. Biological Control and Biochemical Fungicides

Plant tissue is colonized by a wide range of microbes, which may be endophytic (colonizing the inner surface of the plant) or epiphytic (residing on the outer surface of the plant). In many cases endophytic and epiphytic microbes do not cause disease symptoms, However, their interaction with the host plant is not strictly mutualistic and under particular environmental conditions previously mutualistic fungi may become pathogenic [137,138,139]. Nevertheless, many plant-associated microbes play vital roles in promoting plant health, including influencing their hosts’ disease resistance [140]. These resident microbes can antagonize plant pathogens, and thus are a focus of current research on plant disease management [141]. A common mechanism for this antagonism is direct competition for resources, which reduces the availability of both physical space and nutrients in host tissues and, in turn, the opportunity for pathogen infection [142,143]. Plant-associated microbiota can also significantly impact host resistance to pathogen infection through colonization, which triggers the host’s localized defence mechanisms through ISR, resulting in a more rapid induction of the defence response when a plant pathogen subsequently colonizes the host [144]. Furthermore, plant-associated microbes can target and antagonize plant pathogens either directly, by secreting antifungal or antibacterial secondary metabolites that reduce pathogens’ growth, or indirectly, by secreting metabolites that enhance the host’s production of antifungal or antibacterial metabolites [145,146,147]. Some endophytes, such as members of the genus *Trichoderma*, can antagonize plant pathogens directly through mycoparasitism [148,149].

Biocontrol agents are generally plant-associated microbes that have been screened and selected for use in crop systems in order to improve plant fitness, induce plant defence mechanisms, and antagonize pathogens [129,150,151]. Microbes can be beneficial to plant health and disease control through a multitude of mechanisms, and many different biocontrol agents have been developed to protect against pathogens, particularly at the post-harvest stage. In particular, a key characteristic of effective biocontrol agents is to facilitate the exclusion of a pathogen from its ecological niche through effective competition [152]. The current literature on biocontrol focuses heavily on a small number of beneficial genera. For example, species of the soil-borne genus *Trichoderma* are extremely effective in controlling soil-borne and foliar pathogens and have been demonstrated to perform well against a number of pathogenic species, including *Rhizoctonia solani*, *Fusarium oxysporum*, and *Bipolaris sorokinia*. The antifungal properties of these soil-borne species typically arise from a combination of direct competition, mycoparasitism, and the secretion of antifungal compounds [129,149,153,154,155,156]. Members of the genus *Aureobasidium*, most notably *Aureobasidium pullulans*, show promise in controlling post-harvest pathogens, such as *Botrytis cinerea*, *Penicillum expansum*, and *Diplodia seriata* [157,158,159,160,161,162]. Like *Trichoderma*, *A. pullulans* antagonizes pathogens through direct competition for space and resources as well as by the secretion of antimicrobial compounds [163]. Additionally, bacterial genera such as *Pseudomonas*, *Bacillus*, and *Agrobacterium* have demonstrated an excellent ability to suppress economically significant plant pathogens [164,165,166].

Extensive laboratory research has resulted in commercial biocontrol agents being available in varying degrees around the world. *Trichoderma* spp. is the best established biocontrol agent, and several products containing various *Trichoderma* species have been commercialized since the 1970s [167]. Products incorporating fungi such as *Aspergillus flavus* and *A. pullulans*, yeasts such as *Candida oleophila*, and bacteria such as *Pseudomonas*, *Bacillus*, and *Agrobacterium*, have been approved in Europe and the United States [167]. Biocontrol agents are often considered to be less vulnerable to resistance development in pathogen populations, because, unlike synthetic fungicides, they often have multiple modes of action, although this is not always the case, and more research is needed to determine their modes of action [167,168]. Indeed, very few studies have been conducted on the probability of resistance development in fungal pathogens to biological control agents [168]. Despite the potential advantages of biocontrol products, they are not universally preferred to chemical fungicides because their efficacy can vary significantly. Conditions in the field are variable and difficult to predict, and microbial biocontrol agents can behave unreliably in uncontrolled conditions [169].

Mycoviruses also demonstrate remarkable promise as candidates for biocontrol, since mycoviral infections in plant pathogens are associated with a reduction in virulence as a result of the RNA silencing (RNAi) of pathogen and host genes [170]. A major advantage of this form of biocontrol is its ability to spread within a pathogen population following introduction. This spread occurs primarily through fungal hyphal anastomosis (hyphal fusion), a key process required in these pathogens for homeostasis and genetic exchange [171,172]. Upon a mycovirus’ successful infection of a pathogen, the mycovirus will be passed down to the pathogen’s progeny, thereby reducing pathogenicity across multiple generations [173]. To date, numerous mycoviruses capable of inducing hypovirulence in pathogen populations have been identified, including AsHV1, which reduces the virulence and growth of *Alternaria alternata*, and BcPV2, which induces hypovirulence and an absence of conidia in *Botrytis cinerea* [174,175].

In addition to mycoviruses, bacteriophages (viruses that target bacteria) have also been used in the biocontrol of bacterial plant pathogens. Bacteriophages have several advantages, because they are usually genus-specific and able to replicate within their hosts, but do not accumulate in the environment in their hosts’ absence [176]. In some cases, phages can degrade extracellular polysaccharides involved in virulence, as in the case of *Erwinia amylovora* [177]. Bacteriophage-based biocontrol products typically use a combination of different phages to increase the product’s modes of action as well as improve the range of target pathogen genotypes, and to reduce the chance of acquired pathogen resistance [178].

In contrast to biocontrol agents, biochemical fungicides are naturally occurring compounds that can be used to control fungal diseases. Biochemical fungicides can increase a plant’s disease resistance by inducing its defence responses or by directly inhibiting the pathogen. Biochemical fungicides may include compounds screened and selected based on their antimicrobial activity, many derived from biocontrol agents [179] or, alternatively, take the form of whole-cell extracts. Biochemical fungicides have been proven effective against many fungal pathogens. For example, the cell-free extracts of *Pseudomonas* have been shown to inhibit the mycelial growth of *Alternaria alternata* and *Fusarium solani* in vitro, while SH2, an antifungal compound derived from *Streptomyces hydrogenas*, was found to control *Alternaria brassicicola* on radish seeds in vivo [180,181]. Plant extracts are another important source of biochemical fungicides and can inhibit diverse phytopathogenic fungi such as *Botrytis cinerea*, *Fusarium oxysporum*, and *Leptosphaeria sacchari* (syn. *Phoma sorghina*) [182,183]. Similarly, natural oils can have potent antifungal activity, reducing the severity of diseases caused by numerous pathogens, such as *Fusarium oxysporum*, *Fusarium solani*, *Fusarium monliforme*, *Thilaviopsis paradoxa*, *Botryodiplodia theobromae*, and *Rhizoctonia solani* in date palm seedlings; *Phytophthora parasitica* var. *nicotianae* in tobacco; and *Venturia inaequalis* in apple [184,185,186].

Currently, both biocontrol agents and biochemical fungicides are viewed as eco-friendly means of disease control. These methods are typically less phytotoxic than synthetic fungicides and are considered to be a more environmentally friendly method for pathogen inhibition or eradication as they do not leave toxic residues [179,187,188]. As such the identification of biocontrol agents and biochemical fungicides is a significant area of research (Table 2). The continuation of this work will be of vital importance in the future development of disease management strategies.

## 5. RNAi-Based Disease Management

### Small RNA-Based Fungicides

RNA silencing (or RNAi) is a highly conserved mechanism in eukaryotes that allows gene silencing at the transcriptional and post-transcriptional levels by small RNAs (sRNAs). At the transcriptional level, small, double-stranded RNA (dsRNA) is cleaved by the ribonuclease Dicer, and loaded into the RNA-induced initiation of transcriptional gene silencing complex (RITSC), which, utilizing the sRNA as a guide, binds to homologous DNA, leading to the methylation of the heterochromatin in this region and, consequently, gene silencing [224]. A similar mechanism is responsible for post-transcriptional gene regulation, in which small dsRNA, after being cleaved by Dicer, is loaded into the RNA induced-silencing complex (RISC) and guides the latter to complementary RNA targets through base pairing [225]. RISC then “slices” the homologous mRNA directly, leading to the degradation of the mRNA transcript [225,226]. The recognition of the mRNA sequence can also lead to translation repression [225,226,227], since RNAi does not require perfect complementarity. The degree of complementarity between the small RNA guide and its target is a major factor in determining the interference mechanism, with lower degrees of complementarity associated with translational repression and higher ones, with mRNA degradation, although translational repression may still occur in plants under conditions of high complementarity [228,229,230,231]. These mechanisms are originally thought to have evolved primarily as a means of protecting the host from viral infections and transposable elements; however, a body of evidence suggests that hosts and pathogens are engaged in an evolutionary arms race, with a rapid co-evolution of antiviral RNAi genes in hosts and viral RNAi suppressor genes in viruses [232].

In the last decade, RNAi has generated immense scientific interest as a way to achieve effective and eco-friendly disease control in agricultural settings (Table 3). This research can broadly be divided into two separate categories: that focusing on host-induced gene silencing (HIGS), which involves the genetic modification of a host plant to express dsRNA that targets a specific pathogen’s pathogenicity related genes, and that focusing on spray-induced gene silencing (SIGS), which involves the exogenous application of similar dsRNAs or sRNA [233,234].


i.Host-Induced Gene Silencing


RNAi can be triggered by the presence of dsRNA, hairpin or transgenic foreign RNA, or viral dsRNA, culminating in the use of the dsRNA as a template sequence to locate and silence matching foreign dsRNA [233]. Taking advantage of this inherent disease control mechanism, researchers have developed transgenic crops that express dsRNA targeting known pathogens and pests [265]. HIGS offers significant advantages over conventional resistance breeding techniques. For instance, HIGS does not require a pool of established resistance genes. Its only limitation is that the researcher must design the dsRNA to target the relevant pathogenicity genes. The use of HIGS to target the parasitism genes of root-knot nematodes is just one example of the degree of improvement provided by HIGS in fine-tuning disease resistance. For years, efforts to combat root-knot nematode parasitism in many crops were hindered largely by the lack of effective, broad-range resistance genes [266]. However, this problem was overcome by creating transgenic *Arabidopsis* plants that express dsRNA for the parasitism gene *16D10*, which confers resistance against a wide range of root-knot nematode species [267]. Thus, HIGS was able to provide a degree of resistance unattainable by conventional methods, and, in the future, other varieties of vulnerable crops will likely be transformed with *16D10* dsRNA [267].

HIGS techniques show significant promise in the control of fungal pathogens. After confirmation was obtained that dsRNA targeting fungal transcripts of *Blumeria graminis* (the causal agent of powdery mildew) drastically impedes the development of this disease in wheat and barley, transgenic plants were developed that express dsRNA to silence pathogenicity genes. Significant disruption of host–pathogen interactions was demonstrated in these transgenic plants, demonstrating HIG’s potential in controlling fungal pathogens [238]. Since then, HIGS techniques have been applied successfully against the fungal pathogens of a wide variety of crops, including *Verticillium* in tomato and *Magnaporthe oryzae* in rice [244,261].

Despite HIGS’ advantages over conventional breeding methods, this emerging technology has many of the same limitations as conventional techniques, namely that introducing new dsRNA in economically important crops and cultivars is a time-consuming process. In addition, significant opposition to genetic modifications may be present in some jurisdictions. Furthermore, although it may be more difficult for fungi to develop resistance to RNAi, there is significant evidence supporting an evolutionary arms race between fungal pathogens and host plants in regard to RNAi [232]. Therefore, there might be a need to “pyramid” dsRNA to target several relevant (and unrelated) pathogenicity genes [80]. Even with the “pyramiding” of dsRNA, the introduction of resistant cultivars remains a suboptimal option for perennial crops with lengthy life cycles. In these crops, HIGS will likely need to be complemented with conventional spray programs or the new SIGS techniques.


ii.Using Spray-Induced Gene Silencing as the Basis for RNA-based Fungicides


SIGS techniques involve the spray application of exogenous dsRNA or sRNA to plants. The efficacy of this method was first demonstrated in barley, where the application of exogenous dsRNA targeting three ergosterol biosynthesis genes critical to the integrity of fungal cell membranes resulted in the effective inhibition of the fungal pathogen *Fusarium graminearum* [234]. Since then, SIGS has proved effective against a number of fungal pathogens, including *Botrytis cinerea* and *Sclerotinia scletiorum*, in a variety of crop systems, including strawberry, tomato, and *Arabidopsis* [250,256].

While dsRNA was originally thought to be processed directly by the fungal RNAi machinery, recent evidence suggests that SIGS requires the uptake of sprayed dsRNA by the host plant’s stomata [268]. In their study, Biekenkopf et al. demonstrated the processing of dsRNA by host plants and visualized fluorescently labelled dsRNA travelling through the vascular system of plants, indicating its long-distance spreading to distal areas [268]. Significant (~60%) gene silencing of the targeted *Shp* gene was reported even in aphids that fed on unsprayed distal tissues, confirming the travel of dsRNA through the vascular system to tissues [268]. Indeed, the fungal pathogen *Fusarium graminearum* was also impacted by the dsRNA, even in the plant roots [268]. Previously, it had been demonstrated that the use of larger dsRNAs resulted in the decreased inhibition of target mRNA when dsRNAs were applied exogenously, suggesting that the lower efficacy of larger dsRNA in SIGS must be a result of poor uptake by fungi (while this has no relation to the efficacy of HIGS) [251]. However, since more recent evidence suggests that dsRNA uptake and processing by the host plant plays a significant role in the SIGS of target pathogens, it seems that the size of dsRNA may instead inhibit the uptake of these molecules by the plant, a process required in SIGS [268]. Unfortunately, while these studies provide some insight into the uptake and dispersal of dsRNA in host plants, more research is needed to elucidate the exact mechanism for dsRNA uptake in plant tissues and, subsequently, by fungal pathogens.

While the use of SIGS in crop systems is advantageous, it should also come with the caveat that dsRNAs are not continuously transcribed by the crop, as is the case with HIGS. Until recently, this represented a major hurdle in the practical application of SIGS biopesticides, especially since dsRNAs are known to have a short half-life. However, the development of environmentally friendly, non-toxic, degradable clay nanosheets allows the sustained release of dsRNA for up to 30 days after application, drastically increasing the duration of protection offered by a single spray application [269]. Since both the clay nanosheets and dsRNA have been demonstrated to be environmentally non-toxic, SIGS offers significant environmental advantages over traditional chemical fungicides [269]. A recent RT-qPCR method for quantifying dsRNA in agricultural soils demonstrated that exogenous dsRNA dissipated to below-detectable levels within hours of application, providing further evidence that this spray technique may be far more eco-friendly than conventional fungicides, which are known to persist in the environment over a longer time scale [3,270]. In addition, because dsRNAs are designed to target specific pathogens, SIGS has the potential to avoid many of conventional fungicides’ off-target effects, which greatly alter the microbiome of crops [271]. In this regard, the specific design of the dsRNA is vital since RNAi has recently been noted to have off-target effects, such as targeting mRNA with as few as eleven contiguous nucleotides in common with the dsRNA-provided template, thus posing a substantial risk to both human health and the host plant [272,273]. Consequently, extensive research will be required to determine the off-target effects of specific dsRNA on human health, as well as on target crops and their microbiomes. Nevertheless, the potential advantages of SIGS over conventional fungicides are undeniable.

## 6. Future Directions

Despite recent advances in disease control, comparing a given method’s efficacy across different studies can be quite difficult due to differences in crops, cultivars, the timing of treatment, environmental conditions, and targeted pathogens. Although numerous eco-friendly options have been developed to control fungal diseases, many of these are challenging to use effectively in the field. For example, PDEs were initially considered a viable alternative to conventional pesticides, until subsequent field trials demonstrated that PDEs alone were usually less effective than conventional fungicides [274]. Direct comparisons of these control methods will be required in order to determine their relative efficacy under field conditions.

While biochemical fungicides have been a focus of research efforts for decades and are available in commercial formulations, new bio-degradable, environmentally safe products have significant potential and there is a strong need for their development [275]. Many synthetic fungicides, such as strobilurins, have been developed from natural antimicrobial compounds, and the continued identification, isolation, and production of these compounds could provide a pipeline of highly efficient antifungal agents [181,276]. Although the isolation and use of these antifungal compounds may be appealing, the resulting individual compounds are likely to be similar to modern synthetic fungicides in their reliance on a single target site and, therefore, like strobilurins, may be extremely vulnerable to the development of resistance in the targeted pathogens [36]. In contrast, biochemical fungicides produced from the whole-cell extracts of known mutualists, as well as plant extracts and essential oils, appear to be less vulnerable to resistance development due to their potential for multi-site activity. Essential oils and plant extracts may have up to six different modes of action, such as (i) the inhibition of cell wall formation; (ii) the inhibition of ergosterol synthesis; (iii) the inhibition of mitochondrial electron transport; (iv) interference with RNA/DNA and protein synthesis; (v) interference with efflux pumps; and (vi) the inhibition of cell division. On the other hand, substantial losses in the efficacy of plant extracts and essential oils in field treatments have been observed compared to experiments under controlled conditions, most likely due to a decrease in the stability of these products during storage or transport, or under field conditions [277]. However, in recent years, significant advances have been made in biopesticide formulation, and several non-liquid preparations have been developed for insecticidal purposes [278]. Similar advances in essential oil- and plant extract-based biochemical fungicides will improve their efficacy under field conditions and allow for the more widespread adoption of these important technologies, since the formulation and stability of these products remain the greatest obstacles to their effective use.

Although research on RNAi techniques is in its infancy, both SIGS and HIGS have significant potential as eco-friendly disease control methods. In addition to their high potency and reduced environmental impacts, RNAi techniques may also be less vulnerable to resistance development than modern single-site fungicides. While a single mutation can change the conformation of a target enzyme (for example, CYP51 in ergosterol synthesis) and prevent fungicide binding and activity, RNAi mechanisms are not nearly as vulnerable to evasion, since even the imperfect recognition of an mRNA transcript can still lead to translational repression and, thus, deprive the pathogen of a vital protein [228]. Owing to the relatively recent introduction of the HIGS and SIGS techniques, the rate at which fungal pathogens can develop resistance to dsRNA targeting of pathogenicity-related genes has yet to be assessed. Nevertheless, the dsRNA targeting of multiple unique pathogenicity genes, presenting the pathogen with a challenge that would require mutations in multiple loci to overcome, should obviously be investigated. However, the generation of hosts expressing dsRNA is a lengthy and laborious process and would need to be repeated with several dsRNAs unique to each targeted pathogen to ensure enduring broad-spectrum resistance. Considering the strong consumer sentiment against the genetic modification of crops, SIGS will most likely become the preferred method for controlling fungal pathogens through RNAi. Nevertheless, the development of resistant lines with HIGS and, in particular, combining the pyramiding of the expression of unique dsRNA with R-genes and the modification of susceptibility genes could significantly reduce growers’ costs—particularly labour costs—by reducing the need for pesticide applications.

The timing of fungicide applications is usually informed by plant growth stage and/or the predictive modelling of pathogen risk based on weather forecasting, resulting in greater efficiency and reduced fungicide use [279,280,281,282,283]. More recently, assays to detect inoculum levels in the field have been developed, which allow pathogen pressures throughout the growing season to be assessed and the most effective timing for fungicide application to be determined, resulting in a potential reduction in pesticide use and in the rate of fungicide resistance development [284,285]. The monitoring of inoculum levels will be particularly useful in guiding the application of SIGS, since, unlike many chemical fungicides, well-designed dsRNA should be highly specific to the targeted pathogen. The incorporation of environmentally friendly disease control methods in integrated spray programs, informed by predictive modelling and the monitoring of inoculum levels in the field, could significantly reduce our reliance on environmentally toxic chemical fungicides and delay the development of fungicide resistance in economically significant pathogens. For instance, *Colletotrichum acutatum*, a fungal pathogen causing bitter rot of apple, has been demonstrated to overwinter in trees by infecting the outer bud scales asymptomatically; the primary inoculum identified in orchards in early spring is mainly dispersed by the buds opening in the canopy [286]. Therefore, to manage *Colletotrichum acutatum*, early applications of fungicides during bud burst are critical to control the levels of primary inoculum and thus prevent secondary infections later in the growing season [286]. Subsequent to bud break treatments with systemic fungicides, biochemical fungicides or SIGS could be used to control pathogen population levels, reducing the need for additional systemic fungicide applications later in the growing season, delaying the development of fungicide resistance, and reducing the environmental impacts of fungicide use. Weather-based forecasting models for biocontrol and pathogen populations need to be developed to optimize the timing of applications.

Although PDEs and biocontrols are generally outperformed by other disease control methods in comparative assays, these methods are still beneficial in reducing disease under actual field conditions, particularly when used in combination with other techniques. For instance, the co-application of the PDE Actigard^®^ with conventional fungicides significantly increased protection against blue mould of tobacco caused by the oomycete *Peronospora tabacina* compared to the use of fungicides alone [114]. Combining biocontrol agents with chemical and biological plant defence inducers and chemical fungicides has been suggested as a way to reduce fungicide use while still providing effective disease control [287]. In the case of biological PDEs and biocontrol agents, this approach requires compatibility with specific conventional fungicides. However, this is another major advantage of RNAi technologies because RNA constructs, which are designed specifically to target pathogenicity-related genes, are likely to be compatible with unrelated biocontrol agents, enabling their co-application. The successful co-application of biocontrol agents and SIGS would present pathogens with simultaneous multiple challenges, such as RNAi targeting key pathogenicity genes, direct competition for space and resources, the induction of host defences, and direct antagonism. Consequently, at a time when SIGS research is still in its infancy, the investigation of combined treatments with biocontrol agents or PDEs could clearly present a significant opportunity for improved disease management.

## 7. Conclusions

While chemical fungicides have been one of the most effective methods of disease control for decades, restrictions on their use are increasing due to their negative environmental impacts and consequences for animals and human health, as well as the growing threat of fungicide-resistant pathogens. However, the gap left by the elimination of these fungicides cannot be bridged by traditional breeding methods, because the introduction of new resistance genes in plant species is time- and labor-intensive, and pathogenic fungi are able to quickly overcome introduced resistance genes in the field. Consequently, for decades, researchers have been searching for effective and environmentally friendly methods for controlling fungal pathogens.

According to the literature, both biochemical fungicides and RNAi-based techniques are highly effective and have significant advantages over conventional methods in terms of environmental sustainability. The rapid breakdown of RNA in the environment suggests that RNAi techniques will be an eco-friendly alternative to chemical fungicides. In addition, owing to the specificity of RNAi, it can be more easily incorporated in integrated pest management systems than conventional fungicides, leading to combined treatments with biocontrol or PDEs. More research is required in order to determine the relative efficacy of these methods under field conditions and to develop integrated pest management systems that challenge fungal pathogens through a variety of different mechanisms. This will allow a reduction in chemical fungicide use and, therefore, prolong pathogens’ sensitivity to these vital tools while reducing environmental and human exposure. In conclusion, the successful integration of SIGS, biochemical fungicides, biocontrol, and PDEs in conventional spray programs will be beneficial not only to crop health and yield but also to the broader environment and human health, and preserve the efficacy of fungicides by challenging pests with different selection pressures.

## Figures and Tables

**Figure 1 plants-12-03822-f001:**
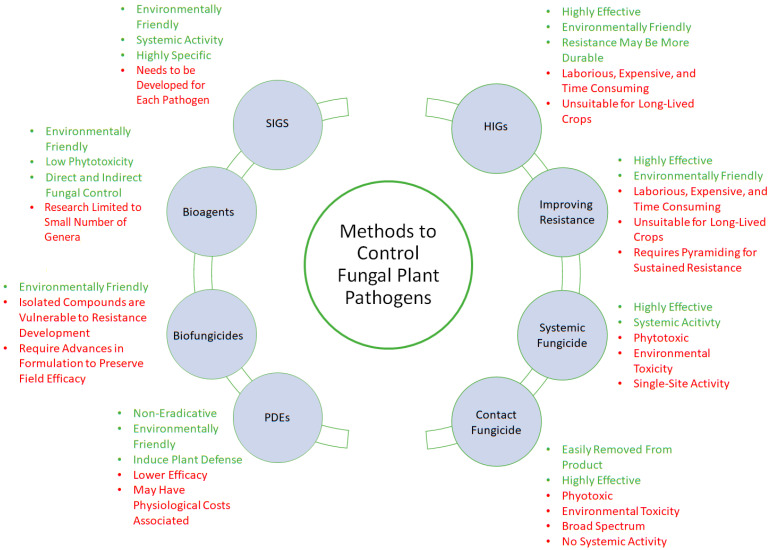
Methods for controlling fungal plant pathogens. The potential advantages (green) and disadvantages (red) inherent in conventional and alternative methods of disease control are also shown.

**Table 1 plants-12-03822-t001:** Examples of the control of fungal pathogens by plant defense elicitors in controlled settings.

Crop	Pathogen	Reference
Apple	*Glomerella cingulate*	[100]
Arabidopsis	*Alternaria alternata*, *Botrytis Cinerea*, *Colletotrichum brassicola*	[115,116]
Broccoli	*Alternaria brassicola*	[117]
Cucumber	*Alternaria cucumerinum*, *Botrytis Cinerea*, *Colletotrichum lagenarium*, *Didymella applanta*, *Fusarium oxysporum*	[118,119]
Carrot	*Alternaria radicina*, *Botrytis Cinerea*	[120]
Tomato	*Alternaria solanii*, *Botrytis Cinerea*, *Verticillium dahliae*	[108,121,122,123]
Strawberry	*Colletotrichum acutatum*	[124]
Cassava	*Colletotrichum gloesporoides*	[125]
Soybean	*Corynespora cassiicola*, *Fusarium oxysporum*	[124,126]
False Brome Grass	*Fusarium graminearum*	[105]
Gooseberry	*Fusarium oxysporum*	[127]
Watermelon	*Fusarium oxysporum*	[128]
Blackberry	*Fusarium oxysporum*	[129]
Potato	*Fusarium solani*	[130]
Sunflower	*Golovinomyces chichoracearum*	[131]
Wild Rocket	*Pletcosphaerella cucumerina*	[132]
Courgette	*Podosphaera xanthii*	[106]
Wheat	*Puccinia triticana*, *Zymoseptoria tritici*	[133,134]
Japonica Rice	*Rhizoctonia solani*	[135]
Hybrid Tea Rose	*Sphaerotheca pannoca*	[136]
Sweet Pepper	*Alternaria solanii*	[122]

**Table 2 plants-12-03822-t002:** Examples of the biological control of fungal pathogens in controlled settings.

Method	Crop	Pathogen	References
Biochemical Fungicide	Astragalus	*Alternaria solani*	[189]
	Banana	*Colletotrichum musae*	[190]
	Barley	*Bipolaris sorokinia*	[154]
	Basil	*Fusarium moniliforme*, *Fusarium oxysporum*, *Fusarium solani*	[191]
	Bean	*Fusarium oxysporum*	[192]
	Carrot	*Alternaria radicina*, *Botrytis Cinerea*	[120]
	Cotton	*Macrophomina phasesolina*, *Fusarium fujikuroi*, *Rhizoctonia solani*	[193]
	Cucumber	*Podosphaera xanthii*, *Podosphaera fusca*	[194,195,196]
	Hybrid Tea Rose	*Sphaerotheca pannoca*	[136]
	Onion	*Alternaria porri*	[197]
	Orange	*Penicillum digitatum*	[198]
	Rapeseed	*Sclerotia sclerotiorum*	[199]
	Rice	*Rhizoctonia solani*	[199,200]
	Strawberry	*Botrytis Cinerea*	[201]
	Sunflower	*Golovinomyces chichoracearum*	[131]
	Tomato	*Aspergillus* sp., *Fusarium* sp., *Fusarium oxysporum*, *Botrytis Cinerea*, *Alternaria solani*	[202,203,204,205,206]
	Wasabi	*Erysiphe cruciferarum*	[207]
	Wheat	*Puccinia triticana*	[133]
Biological Agent	Barley	*Bipolaris sorokinia*	[154]
	Bean	*Rhizoctonia solani*	[208]
	Blackberry	*Fusarium oxysporum*	[129]
	Cotton	*Nigrospora oryzae*	[209]
	Hybrid Tea Rose	*Sphaerotheca pannoca*	[136]
	Peanut	*Sclerotium rolfsii*	[210]
	Pear	*Botryosphaeria dothidea*	[211]
	Rapeseed	*Botrytis cinerea*, *Sclerotinia minor*	[212,213]
	Rice	*Pyricularia oryzae*	[214]
	Soybean	*Rhizoctonia solani*	[208]
	Spinach	*Colletotrichum coccodes*, *Colletotrichum truncatum*, *Colletrichum spinaciae*, *Myrothecium verrucaria*	[215]
	Strawberry	*Botrytis cinerea*	[157]
	Sugar Beet	*Sclerotium rolfsii*	[216]
	Tobacco	*Botrytis cinerea*	[175]
	Tomato	*Alternaria solani*, *Aspergillus* sp., *Fusarium* sp., *Fusarium oxysporum*, *Botrytis cinerea*, *Cladosporium fulvum*	[202,217,218,219,220,221,222,223]
	Wasabi	*Erysiphe cruciferarum*	[207]
	Wild Rocket	*Pletcosphaerella cucumerina*	[132]

**Table 3 plants-12-03822-t003:** Examples of RNAi based control of fungal pathogens in controlled settings.

Method	Crop	Pathogen	References
HIGS	Arabidopsis	*Fusarium graminearum*, *Sclerotinia sclerotiorum*	[235,236,237]
	Barley	*Blumeria graminis*	[238]
	Benthi	*Rhizoctonia solani*, *Verticillium Dahliae*	[239,240]
	Potato	*Botrytis cinerea*	[241]
	Rice	*Magnaporthe oryzae*	[242,243,244]
	Soybean	*Phakopsora pachyrhizi*, *Sclerotinia sclerotiorum*	[245,246]
	Tobacco	*Ciboria shiraiana*	[247]
	Wheat	*puccinia striiformis*	[243,248,249]
SIGS	Arabidopsis	*Sclerotinia sclerotiorum*	[250]
	Barley	*Fusarium graminearum*	[250,251,252,253]
	Citrus	*Penicillium italicum*	[251,252,253,254]
	Cucurbit	*Podosphaera xanthii*	[254,255]
	Grape	*Botrytis cinerea*	[255,256,257]
	Lettuce	*Botrytis cinerea*	[256,257]
	Onion	*Botrytis cinerea*	[256]
	Rapeseed	*Botrytis cinerea*, *Sclerotinia sclerotiorum*	[250,252]
	Rice	*Magnaporthe oryzae*	[244,258]
	Rose	*Botrytis cinerea*	[244,256,258]
	Sativa	*Botrytis cinerea*	[259]
	Soybean	*Phakopsora pachyrhizi*	[245]
	Strawberry	*Botrytis cinerea*, *Botryitis fuckeliana*	[245,256,260]
	Tomato	*Botrytis cinerea*	[256,260]
	Wheat	*Fusarium asiaticum*, *Fusarium culmorum*, *Fusarium graminearum*	[261,262,263,264]

## Data Availability

The paper will be freely available.

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
