# Peer review of "Why Do We Need Alternative Methods for Fungal Disease Management in Plants?"

_plants, 2023, doi:10.3390/plants12223822_

Round 1

Reviewer 1 Report

Comments and Suggestions for Authors

Dear authors

 The manuscript “Why Do We Need Alternative Methods for Fungal Disease Management in Plants. ?” is easy to read and is well written. Presents a careful review with updated references and covering state of the art methods. I believe it has the quality to be published in this journal.

I only  have minor considerations to report:

Introduction

Pag 2. Line 14. Different formatting in the word “specific” as well as in line 34 “the late 19th century

Pag3 : last paragraph drainage [29{Bereswill, 2012 #260]] please correct

Use of plant elicitors

Pag 7, 4th paragraph Arabidopsis is badly written. Please correct. ZhiNenCong (ZNC), an extract of Paecilomyces variotii, stimulated immunity in Aribidopsisthaliana to

Biological Control and Biofungicides

Pag 9 4th paragraph Biofungicides have been proven effective against many fungal pathogens. different formatting

ii. Using Spray-Induced Gene Silencing as the Basis for RNA-based Fungicides

pg 11, 2nd paragraph Fusarium graemarium is missepling.

Conclusions

The figure 1 presented in the end of the manuscript should be referred in the main text. It appears without any contextualization. 

Author Response

please see the attachment, thanks a lot

Reviewer 2 Report

Comments and Suggestions for Authors

1. The author focused to reduce fungal diseases in plants but the contents and arrangement of this review is very poor, looks like a report not a review article.

Therefore i must strongly suggest the author to address the below comments and re-submit it.

2. Please list, biotic and abiotic factors involved and briefly discuss with recent citations.

3. Then Re-write your hypothesis

4. Once you are done with point 2 then must add Tables, and figures with references to support the novelty of your work.

5. The current draft has just 1 flow chart, not enough.

6. Conclusion must show how plant scientists will be benefited from this review.

Author Response

please see the attachment, thanks a lot

Reviewer 3 Report

Comments and Suggestions for Authors

The manuscript: “Why Do We Need Alternative Methods for Fungal Disease Management in Plants?” by Michael Shayne McLaughlin, Maria Roy, Pervaiz A. Abbasi, Odile Carisse, Svetlana Yurgel and Shawkat Ali, provides useful information on possible alternative methods to conventional ones for the control of phytopathogenic fungal microorganisms. The use of elicitors, antagonistic microorganisms, biofungicides and RNAi-Based Disease Management are covered in detail. It is advisable, however, to also consider new agronomic strategies aimed at improving vegetative performance and, therefore, the natural resistance mechanisms of plant hosts, such as, for example, precision agriculture and innovative cultivation methods. Furthermore, the spaces between words, the font size and the names of the species reported in the text (Phytophthora, for example) must be checked.

Author Response

please see the attachment, thanks a lot

Reviewer 4 Report

Comments and Suggestions for Authors

The article entitled "Why Do We Need Alternative Methods for Fungal Disease Management in Plants?" is an excellent review of current methods for the control of  pathogenic fungal. I have to congratulate the authors for the excellent work done, the extensive review, and the well-documented issues.

Please, find attached document with some notes and small comments.

Author Response

please see the attachment, thanks a lot

Round 2

Reviewer 2 Report

Comments and Suggestions for Authors

This version has been improved and accepted for publication.